# A CRISPR/Cas9-Based Assay for High-Throughput Studies of Cancer-Induced Innervation

**DOI:** 10.3390/cancers15072026

**Published:** 2023-03-29

**Authors:** Sapthala Loku Galappaththi, Brenna Katz, Patrick H. Howze, Gregory Hoover, Simon Grelet

**Affiliations:** 1Department of Biochemistry and Molecular Biology, College of Medicine, University of South Alabama, Mobile, AL 36688, USA; 2Mitchell Cancer Institute, The University of South Alabama, Mobile, AL 36604, USA

**Keywords:** cancer innervation, neuronal differentiation, axonogenesis, neurogenesis, β3-tubulin, cancer plasticity, epithelial-mesenchymal transition

## Abstract

**Simple Summary:**

High-throughput methods that accurately monitor cancer-induced neuronal differentiation are a prerequisite to developing genetic and drug screening approaches that may enable the development of new therapeutics targeting cancer innervation. Here we evaluate a fluorescence-based reporter strategy that reports cancer-induced neuronal differentiation in high-throughput settings, allowing insights development into the underlying mechanisms of cancer-induced neuron differentiation and helping to enable therapeutic targeting of cancer innervation.

**Abstract:**

The aggressive nature of certain cancers and their adverse effects on patient outcomes have been linked to cancer innervation, where neurons infiltrate and differentiate within the cancer stroma. Recently we demonstrated how cancer plasticity and TGFβ signaling could promote breast cancer innervation that is associated with increased cancer aggressivity. Despite the promising potential of cancer innervation as a target for anti-cancer therapies, there is currently a significant lack of effective methods to study cancer-induced neuronal differentiation, hindering the development of high-throughput approaches for identifying new targets or pharmacological inhibitors against cancer innervation. To overcome this challenge, we used CRISPR-based endogenous labeling of the neuronal marker β3-tubulin in neuronal precursors to investigate cancer-induced neuronal differentiation in nerve-cancer cocultures and provide a tool that allows for better standardization and reproducibility of studies about cancer-induced innervation. Our approach demonstrated that β3-tubulin gene editing did not affect neuronal behavior and enabled accurate reporting of cancer-induced neuronal differentiation dynamics in high-throughput settings, which makes this approach suitable for screening large cohorts of cells or testing various biological contexts. In a more context-based approach, by combining this method with a cell model of breast cancer epithelial-mesenchymal transition, we revealed the role of cancer cell plasticity in promoting neuronal differentiation, suggesting that cancer innervation represents an underexplored path for epithelial-mesenchymal transition-mediated cancer aggressivity.

## 1. Introduction

The cancer microenvironment is a complex ecosystem composed of malignant cells and non-malignant components, including blood and lymph vessels, endothelial cells, cancer-associated fibroblasts, immune cells, adipocytes, neuronal cells, cytokines, extracellular vesicles, and extracellular matrix [1,2,3,4]. These components play a critical role in determining cancer biology and influencing many aspects of cancer progression, such as development and metastasis [5]. A number of studies have investigated the interactions between cancer cells and non-malignant cells in the cancer microenvironment and their pleiotropic functions driving cancer progression [6]. Understanding these interactions is now considered a promising approach for uncovering potential therapeutic targets or enhancing the efficacy of existing anti-cancer therapies through combination therapy [1]. In particular, the role of neuronal cells in affecting cancer biology has emerged as an exciting area of research, with evidence indicating that the nervous system plays a crucial role in cancer initiation, [7] growth, [8] and metastasis, [8,9] and cancer innervation significantly impact patient overall survival [10,11]. 

The interactions between the nervous system and cancer are reciprocal, and the neuro-epithelial interface established during cancer innervation needs to be further explored to develop new anti-cancer therapies that account for the neuronal component of cancer [3,6,12]. Cancer innervation is composed of two distinguishable processes: perineural invasion (PNI) and cancer neurogenesis (CNG) [13]. Perineural invasion consists in the infiltration of cancer cells around preexisting nerves, which is commonly observed in cancers located in highly innervated tissues such as pancreatic cancer, [14] prostate cancer, [15] head and neck cancer, [13] and colorectal cancer [16]. During perineural invasion, cancer cells establish a new nerve-cancer interface by stimulating neuronal cells, which sprout into the cancer stroma to form a new intratumoral network [12]. Cancer tissue can also induce phenotypic changes in the neurons that innervated the original tissue. For instance, these neurons may undergo a switch from a sensory neuron phenotype to a sympathetic neuron phenotype, further contributing to the complex interplay between cancer and innervation [17]. Cancer neurogenesis consists of the de novo generation of nerves in primary cancer, and recent breakthroughs revealed how neuronal progenitors originating from the central nervous system have the ability to migrate and differentiate into cancer stroma to establish new innervation [18]. Overall, mechanisms in which cancers promote their own innervation remain under investigation. The crosstalk between neurons and cancer cells can occur through direct or indirect interactions and can even be mediated by various elements of the cancer microenvironment [6]. Cancer cells have been observed to produce neurotropic and axon guidance molecules, which promote the growth of intratumoral axons [12,13,15,19,20]. Their functions in cancer biology depend on multiple factors, including the type of cancer, the nature of the involved neurons, and the mechanisms leading to their aberrant expression in cancer cells will require further investigation [21,22,23]. Seminal studies have implicated the role of Semaphorins in cancer-induced neurogenesis and aggressivity [12,15,24], and in our previous study, we identified the TGFβ pathway as a mediator of cancer innervation, promoting the expression of the Semaphorin-4F in breast cancer cells and leading to increased cancer innervation associated with enhanced metastasis [19]. ER-stress has also been recently identified to promote the production of proBDNF in human prostate, colon, and pancreatic cancer and is linked to cancer innervation [25]. Additional studies demonstrated how other axon guidance molecules are involved in cancer innervation, such as Plexin B3 produced by breast cancer cells [26] or EphrinB1 secreted by head and neck squamous cell carcinoma through exosomes [13]. Overall, in the growing field of cancer neuroscience, while several mechanisms have been identified, there is still a need to prospect for the exact mechanisms and how they can vary according to the biological context [12,27]. Importantly, potential therapeutic targets associated with cancer innervation will require further investigation, and the development of new and accurate assays to study their impact on cancer innervation and how they can be disrupted therapeutically is essential. Currently, the methods for studying cancer innervation and cancer-induced neuronal differentiation rely primarily on microscopic approaches, such as immunohistochemistry or immunofluorescence [13,17,18]. In vivo methods for studying neuronal density within tumors typically involve pathology scoring, with intratumoral neuronal density assessed by immunohistochemistry (IHC) combined with stereology techniques [11,15,20]. In contrast, in vitro studies usually rely on measuring changes in neuronal morphology, including neurite length, branches, or number [13,19,25,28]. While effective, these approaches are time-consuming and do not easily provide a comprehensive picture of neuronal differentiation or the intermediate state of differentiation that may occur in the context of cancer. More importantly, these methods do not allow for high-throughput analyses at a single-cell level or the screening of a large cohort of biological conditions, highlighting the need for more efficient and streamlined methods for studying cancer innervation.

Our study explores the potential of endogenous tagging of the general maturation marker β3-tubulin with GFP fluorophore using the CRISPR-Based Homology Independent Targeted Insertion (HITI) approach to study cancer-induced neuronal differentiation that can be applied in vitro and in vivo, and that will enable the development of new high-throughput assays required for investigating cancer innervation. 

## 2. Results

### 2.1. Cancer-Induced Neuronal Precursor Differentiation in Coculture In Vitro 

Previous studies have often exposed PC12 cells to cancer cell supernatants and tracked their morphological change as a reflection of their differentiation to study the effects of cancer on neuronal differentiation. The PC12 line is a widely used neuroblast cell model that was established from a transplantable rat adrenal pheochromocytoma [29]. PC12 cells offer many advantages as they are easily transfectable and have a high yield of CRISPR conversion. Additionally, they display strong responsiveness to differentiation stimuli, such as nerve growth factor (NGF) [30] or cancer cell supernatants [13,15,19], making them an ideal tool for studying neuronal plasticity in vitro [13,19,31]. However, most previous studies using this model in cancer biology tracked the stimulation of PC12 cells by soluble factors and vesicles without considering the direct cell-to-cell interactions established at the neuroepithelial interface of cancer cells.

To study the impact of direct nerve-cancer contacts on neurite formation in PC12 cells, we established a direct coculture system where cancer cells are mixed with PC12 precursors in direct coculture (Figure 1). The PC12 cells are distinguished from the cancer cells by stable and constitutive overexpression of the GFP fluorophore (Figure 1A). Fluorescence microscopy revealed that PC12 cells cultivated alone displayed a round morphology without significant formation of neurites (Figure 1A—left). However, when cocultured with immortalized mouse mammary gland cells (NMuMG), we observed extensive development of neurites extensions, indicating differentiation of neuronal precursors and establishing a new nerve-cancer interface (Figure 1A—right). Low magnification analysis of the coculture showed the overall development of an extended network of neurites across the coculture (Figure 1B).

While the constitutive GFP approach allows for the observation of PC12 precursor differentiation in coculture models through morphology-based measurement, it does not enable rapid, high-throughput quantification of neuronal differentiation. To overcome this limitation, we tested a reporter assay that can be used for the direct assessment of the neuronal differentiation status within the coculture without any morphological-based analysis.

### 2.2. Endogenous Labeling of β3-Tubulin Microtubule Protein in Neuronal Precursors

The CRISPR-associated protein 9 (Cas9) system, guided by a single guide RNA (sgRNA), has emerged as a powerful gene editing tool since its first publication in 2012 [32]. The Cas9-sgRNA complex can induce site-specific double-strand breaks in DNA, which can be repaired via either the non-homologous end joining (NHEJ) or homology-directed repair (HDR) pathway. The NHEJ pathway can result in gene knockout, deletion, inversion, integration, or correction, while the HDR pathway enables targeted gene integration via donor DNA with homologous sequences to a specific chromosomal site [33,34]. We utilized the HITI approach based on the NHEJ repair pathway because this route has been shown to be more efficient in neuronal cells [34]. 

The formation and maintenance of differentiated neuron morphology are essential to their function and stability; these processes rely on the components of the cytoskeleton. These components, consisting of microtubules (neurotubules), intermediate filaments (neurofilaments), and microfilaments (actin filaments), work together to form mature neurons, drive their development, and maintain elongated structure and the unique organization of terminally differentiated neurons [35,36]. Among these cytoskeletal components, the β3-tubulin expression is a widespread, specific, and abundant marker in differentiating and mature neurons [35]. The accumulation of β3-tubulin monomers during neuronal differentiation plays a critical role in sustaining the typical elongated neurite structure of mature neurons, making it a valuable marker of neuronal differentiation, regardless of neuron subtype [37]. Finally, previous neurosciences studies have already demonstrated successful endogenous labeling of β3-tubulin in neuronal cells, making it an ideal target for our study [34,38].

To introduce the GFP sequence into the TUBB3 gene locus with the HITI strategy, we employed the Open Resource for the Application of Neuronal Genome Editing (ORANGE) tool, developed by Willems and colleagues [38]. This tool includes a knock-in library for labeling endogenous proteins, including various neuronal markers such as β3-tubulin. The spCas9, guided by gRNA, created a double-strand break at the TUBB3 gene locus, allowing genomic integration of GFP at the C-terminal end of the β3-tubulin (Figure 2A). A single vector containing spCas9, the gRNA, and the GFP donor DNA sequence flanked by target sequences matching the genomic TUBB3-targeted region was used (Figure 2A). 

After five days of incubation following transfection with the β3-tubulin-GFP knock-in construct, the PC12 cell cultures were subjected to fluorescence-activated cell sorting (FACS) to isolate cells that successfully integrated the GFP. Flow cytometry analysis usually showed a knock-in conversion rate of 1–2% based on assessing GFP^+^ cells (Figure 2B). To generate isogenic cell populations with endogenously integrated GFP to the β3-tubulin monomer, single cells were collected and regrown in 96-well plates (Figure 2B). Once the colonies reached a population of approximately 1000 cells, as determined by an automatic fluorescence microscope plate reader, a clonal GFP+ population was selected and regrown for routine culture. This cell line is referred to as PC12-TUBB3-GFP. 

PC12 cells have been previously shown to undergo differentiation when treated with Nerve Growth Factor (NGF) [39]. Therefore, PC12-TUBB3-GFP cells that had successfully integrated the GFP tag were treated with NGF to maximize the expression of the β3-tubulin-GFP protein. Observation by confocal microscopy showed the expression of a GFP network throughout the cytoplasm of the neuron precursor cells, with notable enrichment in the neurites where β3-tubulin acts as a scaffold (Figure 2C—white arrows). Overall, the GFP signal displayed a tubular network structure, suggesting specific and accurate integration of the fluorophore with the β3-tubulin protein that did not impair polymerization. There was no evidence of aberrant, nuclear, or diffuse fluorophore expression, suggesting an absence of off-target integration and unintended GFP expression directly from the knock-in vector.

When comparing non-treated versus NGF-treated PC12-TUBB3-GFP cells, brightfield microscopy revealed a significant change in the cells’ morphology, confirming that the knock-in approach did not negatively impact their capacity for neurite outgrowth during differentiation (Figure 2D,E, top). Furthermore, fluorescence microscopy imaging of the cells revealed a substantial increase in the GFP signal after differentiation, suggesting that the GFP signal is a reliable indicator of cell differentiation (Figure 2C,D, bottom).

### 2.3. Characterization of the TUBB3-GFP Knock-in Approach

To assess the ability of our GFP reporter to capture intermediate states of neuronal differentiation accurately, a Western blot analysis was conducted on protein lysates extracted from a culture of PC12-TUBB3-GFP cells treated with NGF over a time course of 2 to 8 days (Figure 3A). The results showed a progressive increase in GFP expression, with a signal observed at around ~82 kDa, matching the size of the combined GFP (27 kDa) and β3-tubulin protein (55 kDa). The progressive induction of neuronal differentiation and GFP expression upon NGF treatment was further validated through conventional fluorescence microscopy, which revealed an incremental increase in signal over time (Figure 3B). To evaluate the sensitivity of our reporter assay in a high-throughput setup, we used flow cytometry to monitor neuronal differentiation in PC12-TUBB3-GFP cells stimulated with NGF over a period of 0 to 10 days, using GFP analysis. This approach provided a rapid, in-depth visualization of the intermediate differentiation states of PC12-TUBB3-GFP cells, not only at the single-cell level but also for a large population of cells (Figure 3C,D) and was also observed in the N2A-TUBB3-GFP cell model whose differentiation occurred in the course of 5 days (Appendix A).

To assess the adaptability of the GFP knock-in approach to other neuronal differentiation models, we also established a TUBB3-GFP knock-in in Neuro-2a (N2A) cells and established the N2A-TUBB3-GFP cell line (Figure 4). Neuro-2a cells are derived from mouse neuronal crest and have been widely used in studying neuronal differentiation, axonal growth, and signaling pathways [40]. Both PC12 and N2A models showed a consistent, similar pattern of increase in TUBB3-GFP expression after 5 days of NGF stimulation, as analyzed by flow cytometry (Figure 4A). This indicated that the CRISPR-based endogenous labeling approach could be applied to other rodent species to reflect the differentiation of their neuronal precursors. 

To determine if endogenous labeling of β3-tubulin had any unintended effects on PC12 and Neuro-2A cell biology, we evaluated the GFP knock-in impact on cell growth and metabolic activity. These are two critical parameters of neuronal precursor biology that can be affected by stress [41,42]. A time course cell proliferation and viability assay using direct cell counting (Figure 4B) and a WST-1 metabolic activity assay (Figure 4C) showed no significant difference between the wild-type and GFP knock-in cell populations. This suggests that the endogenous labeling of TUBB3 did not significantly impact the normal biology of PC12 and Neuro-2A neuronal precursors and did not trigger stress promoting their differentiation.

### 2.4. Validation of the Reporter Approach for High-Throughput Study of Cancer-Induced Neuronal Differentiation in Direct Coculture In Vitro

To evaluate the efficacy of the model in studying cancer-induced neuronal differentiation, we first established a model of breast cancer plasticity through epithelial-mesenchymal transition (EMT). The murine mammary gland (NMuMG) epithelial cells are widely used to study plasticity-mediated cancer progression [43,44,45]. The NMuMG cell cultures obtained from ATCC generate epithelial cell cultures but harbor a small subpopulation of cells possessing fibroblast-shaped appearances (Figure 5A—red arrow). In the past, the heterogeneity of NMuMG cells was leveraged to study the function of E-cadherin in EMT [44,46]. Using the FACS single-cell seeding approach, we generated isogenic subcultures of NMuMG cells and isolated epithelial and mesenchymal subcultures (Figure 5—NMuMG-E; NMuMG-M). Western blot of the two selected clones with representative epithelial (NMuMG-E) or mesenchymal phenotypes (NMuMG-M) showed that NMuMG-M cells had a strong increase in the expression of mesenchymal markers fibronectin, N-cadherin, Slug, and loss of the epithelial marker E-cadherin (Figure 5B). To verify the functional relevance of NMuMG E/M in modeling cell EMT-related plasticity, wound healing assays were performed, and cell velocity was quantified with time-lapse microscopy (Figure 5C,D). We observed significantly increased migratory properties in NMuMG-M compared to NMuMG-E cells (Figure 5C,D), confirming the plasticity-mediated increased aggressivity of NMuMG-M cells. In our previous study, we found that silencing hnRNPE1 or treating cells with TGFβ triggers breast cancer plasticity and increases expression of the Sema4F axon guidance molecule, leading to increased neurite outgrowth in PC12 and Neuro-2a cells and increased cancer innervation in a xenograft model in vivo [19].

The NMuMG-E/M model of breast cancer plasticity provides a unique advantage for studying cancer-induced neuronal differentiation in direct coculture, as it avoids the use of cytokines such as TGFβ, EGF, FGF, or HGF to trigger EMT, which likely could directly impact neuronal cells if applied directly to the coculture [48,49]. This approach also eliminates the need for genetic modification of the cells, such as the ectopic expression of Snail or Twist, which have been shown to be successful in inducing EMT in immortalized mammary epithelial cells, such as HMLE cells [50].

Overall, our reporter strategy aimed to investigate the relationship between cancer plasticity and cancer neurogenesis using a high-throughput setup. 

### 2.5. Breast Cancer Plasticity Controls Neuronal Precursors Differentiation

To explore the link between cancer plasticity and cancer innervation beyond plasticity solely mediated by growth factors, we mixed PC12-TUBB3-GFP cells with either epithelial NMuMG-E cells or mesenchymal NMuMG-M cells. The differentiation of PC12 cells was analyzed by measuring the β3-tubulin-GFP signal. Fluorescence microscopy observation revealed that PC12-TUBB3-GFP exposed to NMuMG-M cells acquired increased β3-tubulin-GFP expression compared to those exposed to NMuMG-E cells (Figure 5E) and consistently exhibited increased changes observed in their morphology, increased neurite outgrowth extension.

In-depth assaying of plasticity-mediated neuronal differentiation was then performed with two complementary high-throughput approaches. In the first setup, we used flow cytometry to screen a large number of neuronal precursors and determine their level of differentiation individually (Figure 5F). This was accomplished by analyzing PC12-TUBB3-GFP cells in a coculture with NMuMG-E or NMuMG-M, as shown in Figure 5F. The flow setup can rapidly assess the differentiation of hundreds of thousands to millions of neuronal precursors in the coculture. These data can therefore be used for building robust statistical values required to measure subtle changes in neuronal differentiation or to precisely explore the diversity in the response intensity of neuronal cells in a given biological context.

In the second setup, we analyzed cocultures established in 96-well plates and measured them using automated fluorimeter readings, as depicted in Figure 5G. The readings from these cocultures provided a bulk cell type analysis, which required additional normalization due to the decrease in neuronal proliferation rates that typically occurs during differentiation [51]. As a result, an increase in differentiation, indicated by an increase in GFP signal per cell, could likely be associated with a decrease in the number of neuronal cells within the coculture. This led to a decrease in the overall GFP signal, even though the neuronal cells exhibited a more mature phenotype. To address this limitation, we added a constitutive mCherry expression to the PC12-TUBB3-GFP cells (PC12-TUBB3-GFP-mCherry) to normalize these data. The GFP/mCherry signal ratio, which represented the overall differentiation state of the neuronal precursors in the coculture, was then examined as a tangible score to report neuronal differentiation in the coculture. This approach helped to overcome the limitation of decreased neuronal proliferation rates and provided an accurate measurement of neuronal differentiation.

Both experimental setups confirmed a significant increase in the differentiation of PC12-TUBB3-GFP and PC12-TUBB3-GFP-mCherry cells while in coculture compared to those cultivated alone (Figure 5F,G). Notably, the differentiation state was significantly increased for neurons exposed to NMuMG-M cells than those exposed to NMuMG-E cells, supporting the role of cancer plasticity in breast cancer innervation. Further evidence of the response of N2A-TUBB3-GFP cells to NMUMG-M cells was observed by flow cytometry analysis of the coculture (Appendix A). Although NMuMG cells represent a model of choice to study breast cancer plasticity, these non-transformed mammary cells do not originate from cancer. To address this, we also tested our approach using the 4T1 breast cancer cells, a well-established mouse model for studying breast cancer metastasis, and observed strong differentiation of the PC12TUBB3-GFP cells mixed with 4T1 (Appendix A).

Overall, our experimental methods and the creation of a breast cancer plasticity model validated the TUBB3-GFP knock-in reporter strategy as an effective tool for investigating the connection between nerves and cancer and could be applied in vitro or in vivo. The approach we developed streamlines the collection and measurement of numerous cells or coculture conditions, allowing for high-resolution and high-sensitivity analysis of the nerve-cancer crosstalk.

## 3. Discussion 

The findings from various studies indicate that the nervous system can play a role in the initiation, growth, metastasis, patient survival, and cancer recurrence rate [4,6,12]. Despite significant efforts by the scientific community in this field, there is still much to be learned about cancer neuroscience. Cancer innervation can occur through axonogenesis, where axons sprout from preexisting nerves into cancer, or through the differentiation of neuron progenitors from the central nervous system [12,15,18]. In both cases, the development of neurite outgrowth is a prerequisite to developing a new intratumoral innervation network. However, the underlining mechanisms of cancer-induced neurogenesis remain largely unknown. Therefore, there is a pressing need to understand the mechanisms of the nerve-cancer crosstalk and to find molecules or therapeutic strategies capable of blocking cancer innervation that is associated with cancer aggravation. Progress in this direction will be made by deploying the screening approaches that will allow for identifying the genes involved in establishing the nerve-cancer interface and through prospecting pharmacological inhibitors of the nerve-cancer crosstalk.

A major challenge in studying cancer innervation is the lack of reliable assays for measuring cancer-induced innervation. In this study, we have developed a precise assay to quantify cancer-induced axonogenesis, which is a major step of cancer innervation [12] Because our approach can be applied to screen a large number of coculture conditions or to screen a large cohort of cells out of nerve-cancer cocultures, it is suitable to develop drug screening or genome-wide library screenings, respectively, and will likely help to undermine the genes, genetic factors, or molecules that represent therapeutic targets against cancer innervation. Interestingly, we observed how cancer cells in coculture could show even a higher effect in neuronal precursor differentiation compared to the NGF stimulation alone (Figure 5F). We are currently investigating the exact mechanism leading to the NMUMG-M-dependent neuronal differentiation, and previous reports have already demonstrated how cancer cells can promote neuronal precursor differentiation through various routes [12,13,15,17,19,25,26,27]. We can speculate that the NMuMG-M cells’ expression of several specific axon guidance molecules triggers a strong response in PC12 cells. In addition, because we provide a method that allows direct coculture of neuronal precursors with cancer cells, the continuous exposure to secreted factors, rather than discrete NGF addition, may sustain a more robust and continuous response. Additionally, direct cell-cell contact may have a more profound impact on triggering neuronal precursor differentiation, as it allows activation of direct cell-cell signaling involving membrane-localized molecules such as plexin/semaphorin. Finally, we hypothesize about a synergistic effect of having multiple molecules and signaling involved simultaneously, as NMuMG-M cells may produce multiple axon guidance molecules promoting differentiation through multiple pathways that together accelerate the differentiation of PC12, while the addition of NGF to PC12 may remain limited through a saturation effect.

Beyond the field of nerve-cancer crosstalk, this method can also be used to assess how non-malignant cells in the cancer microenvironment affect neuronal differentiation and cancer innervation. In this study, beyond the cancer context, we successfully applied our method to compare the differentiation dynamics of neuronal differentiation models PC12 (Figure 3C,D) and Neuro2A cells (Appendix A). Our time-course experiments highlighted differences in response dynamics between these models. While PC12 differentiation occurred over 10 days, the 10 days observation timeframe was not applicable to the N2A model due to distinct kinetic differences characterized by an accelerated reaction of the Neuro2A model. Indeed, following 5 days of NGF treatment, N2A cell proliferation was drastically reduced, and the cells that underwent terminal differentiation eventually died, leading to the loss of the GFP signal. Therefore, flow cytometry analysis provided insufficient data for reliable interpretation at exposure points beyond 5 days. We could hypothesize that N2A cells’ heightened response to NGF could be due to a larger expression of NGF receptors, and we also speculate that this is due to a greater NGF-to-cell treatment ratio because N2A cells had to be seeded at lower confluence as they tend to proliferate more significantly than the PC12 cells. To enable a more gradual differentiation process for extended research in models having strong differentiation responses, such as the N2A model in our context, we, therefore, recommend reducing the NGF concentration.

Though it was not tested in the presented study, successful transfer and adaptation of this strategy into in vivo experimentations is expected and will allow the development of new studies about the nerve-cancer crosstalk in preclinical models of cancer progression.

Finally, our study also supports the role of cancer cell plasticity in promoting cancer innervation. Previous studies have shown that cancer plasticity can contribute to cancer progression through various mechanisms [52,53,54], and we suggest that cancer innervation may represent an additional and underexplored outcome that is promoted by cancer plasticity, which could explain its function in promoting cancer aggressiveness. We believe that further studies exploring this direction will be necessary in the future.

## 4. Methods

### 4.1. Cell Culture and Reagents

The cell lines used in this study, including PC12, Neuro2A, 4T1 and NMuMG, were obtained from the American Type Culture Collection and cultured in high glucose DMEM (Life Technologies, Carlsbad, CA, USA #11995065) supplemented with 5% fetal bovine serum (Fisher Scientific, Waltham, MA, USA #FB12999102), 5% calf serum (Fisher Scientific, #SH3007303HI), and 1% antibiotic/antimycotic solution (penicillin G, streptomycin, and amphotericin B; Life Technologies, #15240062). The cells were maintained in a 37 °C, 5% CO_2_ incubator. Puromycin (#A11138), blasticidin (#ant-bl), and Zeocin (#ant-zn) were purchased from InvivoGen, San Diego, CA, USA. For the experiments involving PC12-TUBB3-GFP cells, the cells were exposed to 50 ng.mL^−1^ nerve growth factor (NGF; Thermofisher, Waltham, MA, USA #A42578) for varying durations up to 10 days. The GFP fluorescence was then measured to assess the level of β3-tubulin protein expression using the various methods presented in the manuscript and detailed below.

### 4.2. Generation of PC12-GFP and PC12 mCherry

PC12-GFP and mCherry-expressing cancer cells were obtained by lentiviral transduction. We first seeded 1 × 10^5^ cells into 6-well plates and allowed them to adhere for 24 h before adding the viral particles. We produced the lentiviral particles in our lab using lentiviral constructs for GFP (Addgene, Watertown, MA, USA #17448—pLenti CMV GFP, a gift from Drs. Eric Campeau and Paul Kaufman) and mCherry (Addgene #131505—pFUGW mCherry-KASH, a gift from Dr. Harold MacGillavry). We used the given lentiviral constructs with psPAX2 and pMD2.G packaging plasmids (Addgene #12260 and addgene #12259, a gift from Dr. Didier Trono) in co-transfection using Lipofectamine 3000 to produce the lentiviral particles. After overnight incubation, we changed the medium to fresh culture medium and collected the virus at 24 and 48 h, and filtered it through a 0.45-μm sterile filter. To transduce the cells, we mixed 1 mL of lentiviral particles with polybrene (2 µg/mL; EMDMillipore, TR-1003) and added them to the cells. We incubated the cells at 37 °C overnight and then replaced the medium with a fresh one. To create stable cell lines, we cultured the cells for 48 h at 37 °C before selecting them with antibiotics for 1–2 weeks. For transient expression experiments, we cultured the cells for at least 96 h but no more than two weeks at 37 °C before experimental analysis. In all cases, we transduced the lentiviral particles using a 1:5–1:2 ratio of virus-containing media to culture media with 8 μg/mL polybrene overnight (EMDMillipore, Temecula, MA, USA, TR-1003).

### 4.3. Generation of PC12-tubb3-GFP/ Neuro2A-tubb3-GFP Cell Lines

To generate PC12-tubb3-GFP and Neuro2A-tubb3-GFP cell lines, the cells were first plated and maintained in 6-well tissue culture dishes at a density of 60–80% confluency. The media was then replaced with DMEM containing 5% fetal bovine serum and 5% bovine calf serum (no antibiotics). A transfection mixture was prepared in a tissue culture hood, consisting of 5 µg of pORANGE Tubb3-GFP (Plasmid #131497—Addgene, a gift from Dr. Harold MacGillavry), 15 µL of FuGENE6 (ProMega, Madison, WI, USA, E2311), and 77 µL DMEM serum-free media. The mixture was added to the cells and incubated for 3 days. The cells were then trypsinized, and the supernatant was removed after spinning down with 2 mL of DMEM media. GFP-positive cells were sorted using flow cytometry.

### 4.4. Flow Cytometry

We used a BD Biosciences FACS Aria II cell sorter with 488 nm laser excitation and 525/30BP detection for GFP, as well as 561 nm laser excitation and 610/20BP detection for the mCherry fluorophore. Typically, 5000 cells were sorted into 6-well plates containing 2 mL DMEM supplemented with 10% FBS and were cultured for at least 96–120 h before subsequent use.

### 4.5. Western Blot Analysis

For Western blot analysis, 2 to 5 × 10^6^ cells were lysed in 300–500 μL of lysis buffer (20 mM Tris, pH 7.4, 1% Triton X-100, 10% glycerol, 137 mM NaCl, 2 mM EDTA, 1 mM Na_3_VO_4_, and protease inhibitors), and whole cell lysates were clarified by centrifugation at 4 °C for 10 min in an accuSpin Micro 17R (Fisher Scientific) microcentrifuge at maximum speed. Protein lysates were separated on 10% acrylamide protean III mini gels and transferred to an immobilon-P membrane (EMDMillipore). After blocking the membrane in wash buffer (PBS containing 0.1% Tween 20) with 5% nonfat dry milk for 1 h, the membrane was incubated overnight with primary antibodies, including a mouse monoclonal GFP antibody (B-2—Santa Cruz, Dallas, TX, USA #sc-9996; 1/200 dilution), a mouse monoclonal antibody against beta3-tubulin (2G10—Abcam, Waltham, MA, USA, #78078; 1/2000 dilution), a recombinant rabbit monoclonal anti-Sox2 antibody (EPR3131—Abcam #92494; 1/5000 dilution), a rabbit monoclonal anti-vimentin (#D21H3—Cell signaling technology, Danvers, MA, USA; 1/500 dilution), a purified mouse anti-fibronectin (BD transduction #610077; 1/1000 dilution), a mouse monoclonal anti GAPDH antibody (6C5—Santa Cruz #sc-32233; 1/10000 dilution) or a mouse monoclonal anti-beta-actin antibody (C4—Santa Cruz #sc-47778; 1/1000 dilution), all diluted in the same blocking buffer. After extensive washing, the blot was incubated with a secondary antibody for 1 h in a blocking buffer, washed, and processed using the ECL+ ChemiDoc^TM^ imaging system (Bio-Rad, Hercules, CA, USA).

### 4.6. Direct Counting Cell Proliferation Assay

PC12-WT, PC12-TUBB3-GFP, N2A-WT, and N2A-TUBB3-GFP cells were seeded in 96-well plates. Their cell counts were measured at hours ~8, 16, 32, 64, 128, 256, and 512 using the Celigo Image Cytometer (Nexcelom Bioscience, Lawrence, MA, USA) with Hoechst 33342 (InvivoGen #H1399) and Propidium Iodide (InvivoGen, #P1304MP) staining to measure cell viability.

### 4.7. WST1 Metabolic Activity Assay

PC12-WT, PC12-TUBB3-GFP, N2A-WT, and N2A-TUBB3-GFP cells were subjected to WST-1 assay (Sigma-Aldrich Inc, Burlington, MA, USA, #5015944001) by adding the assay reagent to the cell culture media according to the manufacturer’s instructions and incubated for 4 hrs. The absorbance at 440 nm was measured using a 96-well BioTek, Winooski, VT, USA, Cytation 7 cell imaging multimode reader and the associated Gen5 software (Agilent Technologies, Santa Clara, CA, USA) to quantify the amount of formazan dye produced.

### 4.8. Migration Assay

NMuMG cells were seeded into a 12-well tissue culture plate using an Ibidi well silicone insert to create a defined cell-free gap. After 24 h, the insert was removed, and time-lapse microscopy was used to trace cell migration in a BZX-800 automated microscope (Keyence, Osaka, Japan).

### 4.9. Fluorimeter and Microscopy Analysis

The fluorescence of the coculture was measured using an Infinite M1000Pro automatic fluorescence plate reader (Tecan, Männedorf, Switzerland), capturing both mCherry and GFP channel fluorescence, with ratios calculated after background subtraction. Microscopy analysis was conducted using either the BZX-800 automatic fluorescence microscope (Keyence) or the Nikon A1r inverted confocal microscope (Nikon, Tokyo, Japan).

### 4.10. Statistical Analysis

Student *t*-test and ANOVA analyses were used for statistical analysis, with *p*-values indicated by asterisks in the legends or as stated in the figures. GraphPad Prism v9.5.0 was used for data analysis.

## 5. Conclusions

Here, we have successfully developed a fluorescence-based reporter strategy that can report cancer-induced neuronal differentiation in high-throughput settings. Our aim was to develop an assay that would streamline the study of cancer-induced neuronal differentiation, allowing for better normalization and comparison of experiments related to this phenomenon and allowing for the direct co-culture of neuron precursors with cancer cells. We hope that this method will enable the development of a better understanding of the underlying mechanisms of cancer-induced neuron differentiation and the therapeutic targeting of cancer innervation.

## Figures and Tables

**Figure 1 cancers-15-02026-f001:**
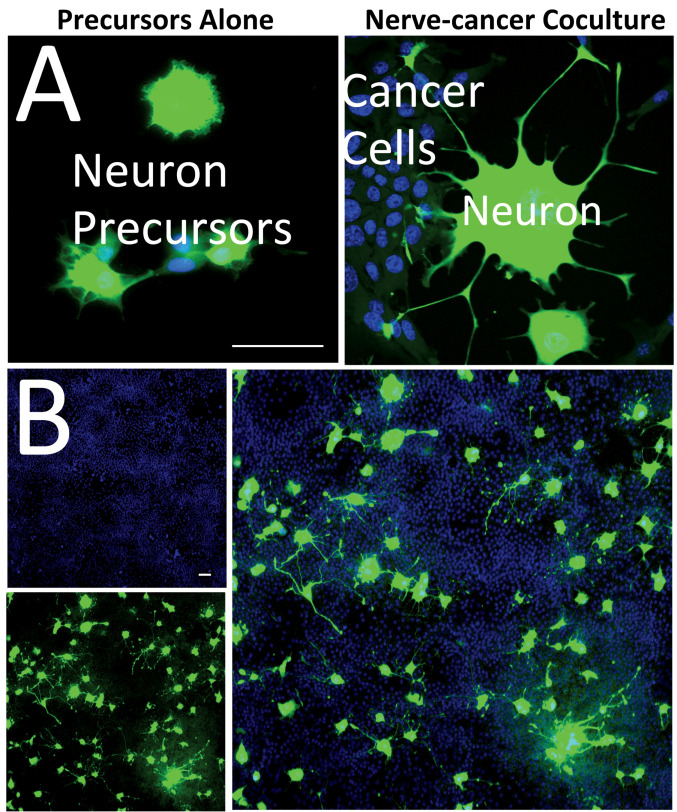
Cancer-induced neuronal precursor differentiation. (**A**) Fluorescence microscopy analysis of PC12 neuronal precursors in direct coculture (nerve-cancer coculture) with breast immortalized cells (NMuMG—Normal Murine Mammary Gland) demonstrates the development of axons and dendrite protrusions that come into contact with the cancer cells (**right**). PC12 cells are distinguished from NMuMG cells by stable, constitutive expression of the GFP fluorophore (green channel), while all cell nuclei of the culture are stained with DAPI (blue channel). PC12 cells cultivated alone (precursors alone) show only a low development of neurites (**left**). (**B**) At low magnification, an extended network of neurites is visible after five days of coculture, suggesting the formation of a new nerve-cancer interface. While the constitutive GFP approach allows for the observation of PC12 precursor differentiation in coculture models by morphology-based measurement of their degree of differentiation, it does not permit rapid, high-throughput quantification of neuronal differentiation and makes it difficult to assess the intermediate states of differentiation. Scale bars: 50 µm.

**Figure 2 cancers-15-02026-f002:**
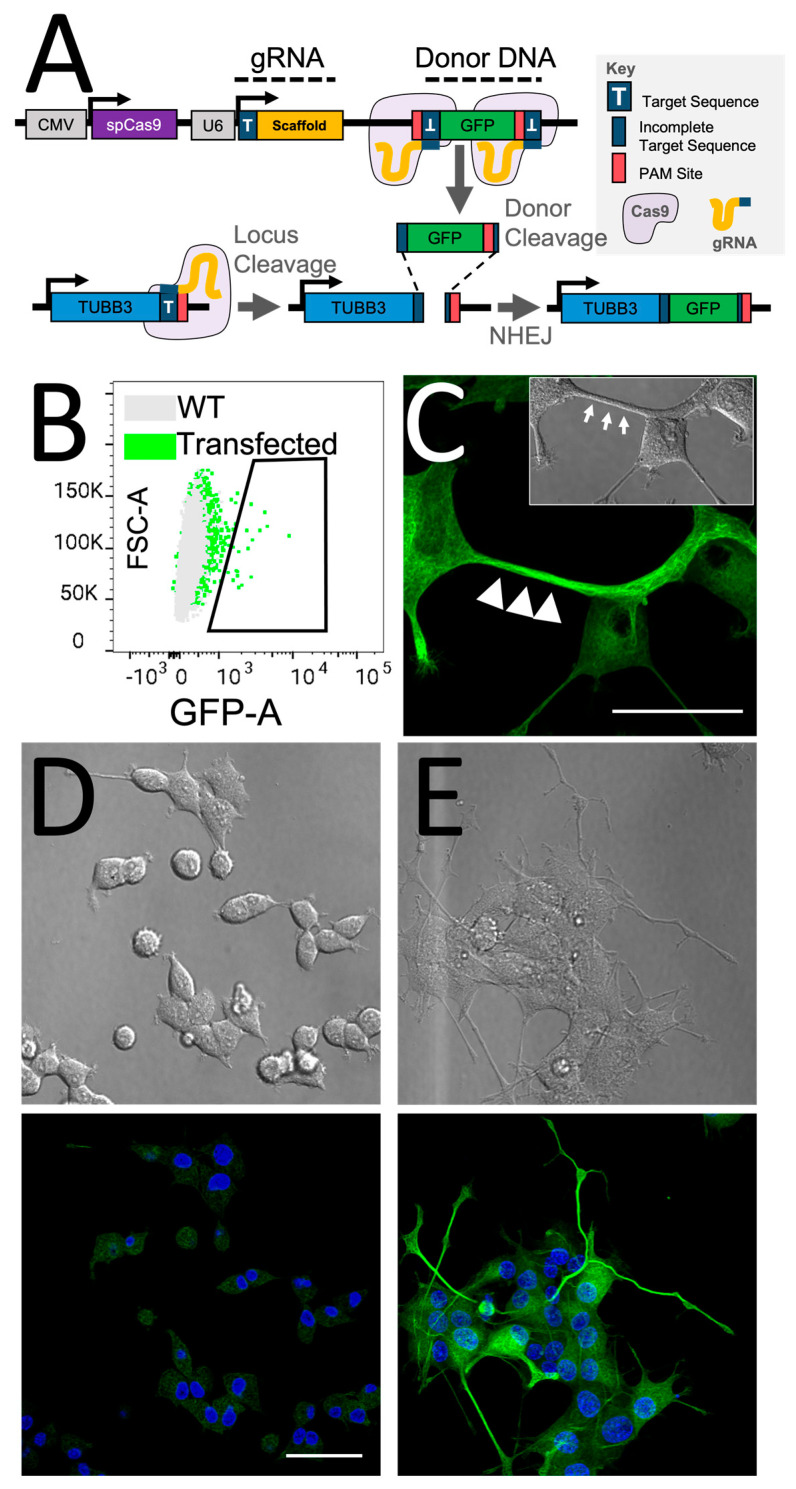
CRISPR/Cas9-based HITI approach for experimental labeling of endogenous β3-tubulin. (**A**) To generate a culture of PC12-TUBB3-GFP cells, neuronal PC12 precursors were transfected with the vector containing Cas9, sgRNA, and the GFP reporter donor sequence, which integrates into the β3-tubulin locus. We used a single vector that contains spCas9 controlled by a constitutive CMV promoter, the gRNA controlled by a U6 promoter, and the GFP donor DNA sequence flanked by target sequences that match the genomic TUBB3-targeted region. The gRNA creates a double-strand break and removes the GFP donor DNA from the plasmid for genomic integration of GFP at the β3-tubulin locus target sequence. The correct orientation of GFP is ensured by the reverse orientation of the target sequence sites and protospacer-adjacent motif (PAM) compared to the TUBB3 locus. This design allows Cas9 to self-correct GFP its orientation if integration occurs backward. (**B**) Strategy to isolate PC12 culture transfected with the knock-in vector. Cells were sorted using fluorescence-activated single-cell sorting (FACS) to select cells that acquired GFP fluorescence signal as presented in the dot plot (transfected). Wild-types (WT) cells served as control. GFP^+^ cells were collected and regrown in 96-well plates to generate isogenic cultures. Cells were then extracted from the well to regenerate the desired culture. (**C**) Confocal microscopy observation of PC12-TUBB3-GFP culture treated with NGF to induce differentiation and the development of extended neurites reveals the expression of a GFP network throughout the cytoplasm of the neuron precursor cells, with particular enrichment within the neurites where β3-tubulin acts as a scaffold (white arrows). The GFP signal reveals a typical and expected tubular network structure specific to tubulin networks such as the targeted β3-tubulin. (**D**,**E**) Fluorescence microscopy imaging confirms increased GFP signal in PC12-TUBB3-GFP neuronal precursors after their differentiation. The top pictures show brightfield imaging confirming the change in neuronal morphology between untreated cells (**D**) and cells treated with NGF at 50 ng/mL for 5 days to induce differentiation (**E**). The bottom picture highlights GFP expression in these cells and the significant increase in signal, validating the reporter’s ability to report neuronal differentiation. Blue: DAPI staining of the nucleus. Green: TUBB3-GFP. Scale bars: 50 µm.

**Figure 3 cancers-15-02026-f003:**
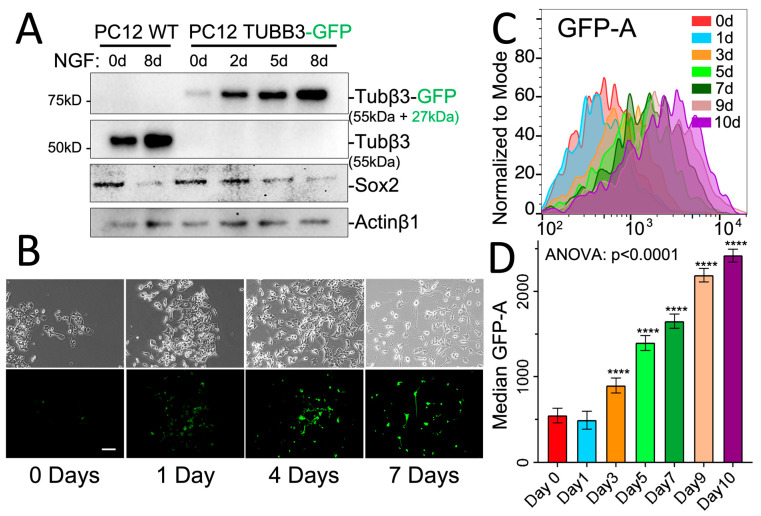
Analysis of intermediate states of neuronal differentiation using the GFP reporter. (**A**) Western blot analysis was performed on protein lysates extracted from a culture of PC12 cells treated over a time course of 2 to 8 days (Original Western blot, see Appendix A). Results showed a progressive increase in GFP expression (Tubβ3-GFP: hybridization with GFP antibody), with a signal observed at around 82 kDa corresponding to the size of the combined GFP and β3-tubulin proteins. Sox2, a marker of neuronal stem cells, was used to indicate neuronal differentiation occurring during the time course. Actinβ1 was used as a loading control. (**B**) The progressive induction of neuronal differentiation and GFP expression was further validated through conventional fluorescence microscopy observation, which revealed an incremental increase in fluorescence signal over time. (**C**) Flow cytometry was used to assess the sensitivity of our approach in a high-throughput setup. We accurately monitored the time response of the neuronal differentiation process in PC12 cell cultures that were stimulated with NGF (100 ng/mL) for 0 to 10 days, providing an in-depth visualization of the intermediate states of differentiation at a single cell level and in a large population of cells. (**D**) The results obtained from the flow cytometry analysis were quantified, and data are expressed as median signal +/− coefficient of variation (CV). Statistics: Ordinary one-way ANOVA and Tukey’s multiple comparison *t*-tests. **** *p* < 0.0001. Scale bar: 50 µm.

**Figure 4 cancers-15-02026-f004:**
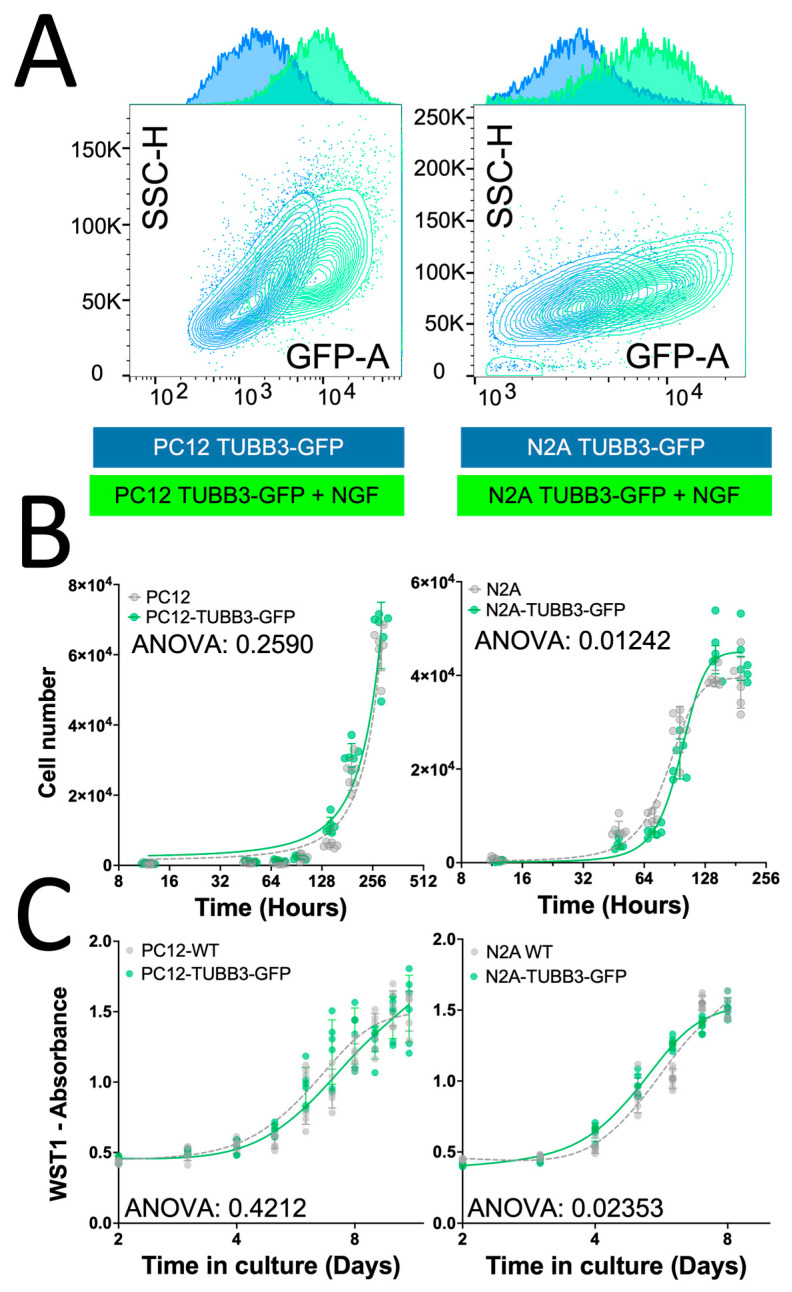
Endogenous tagging of TUBB3 does not compromise cell integrity. (**A**) Endogenous tagging of β3-tubulin was performed in rat-derived PC12 (**left**) and mouse-derived Neuro2A (**right**) cell lines. Both cell lines showed a consistent increase in expression of β3-tubulin-GFP after NGF stimulation (100 ng/mL) as analyzed by flow cytometry. (**B**) Endogenous labeling of β3-tubulin does not compromise cell proliferation of either PC12 (**left**) or Neuro-2A (**right**) neuronal precursors, as observed by a time course cell viability assay using direct counting of Hoechst/PI staining of the cell culture between 0 and 10 days of culture. We observe no statistically significant difference in the rate of cell proliferation between cells that have undergone endogenous labeling and control cells, providing evidence that the CRISPR-mediated labeling approach did not compromise cell viability. (**C**) WST-1 assay does not reveal any changes in cell metabolic activity after endogenous tagging of β3-tubulin in either PC12 (**left**) or Neuro-2A (**right**) cell lines. We observe no statistically significant difference in the cell metabolic activity between cells that have undergone endogenous labeling and control cells, providing evidence that the CRISPR-mediated labeling approach did not affect the cell’s overall differentiation or viability. Statistics: two-way ANOVA.

**Figure 5 cancers-15-02026-f005:**
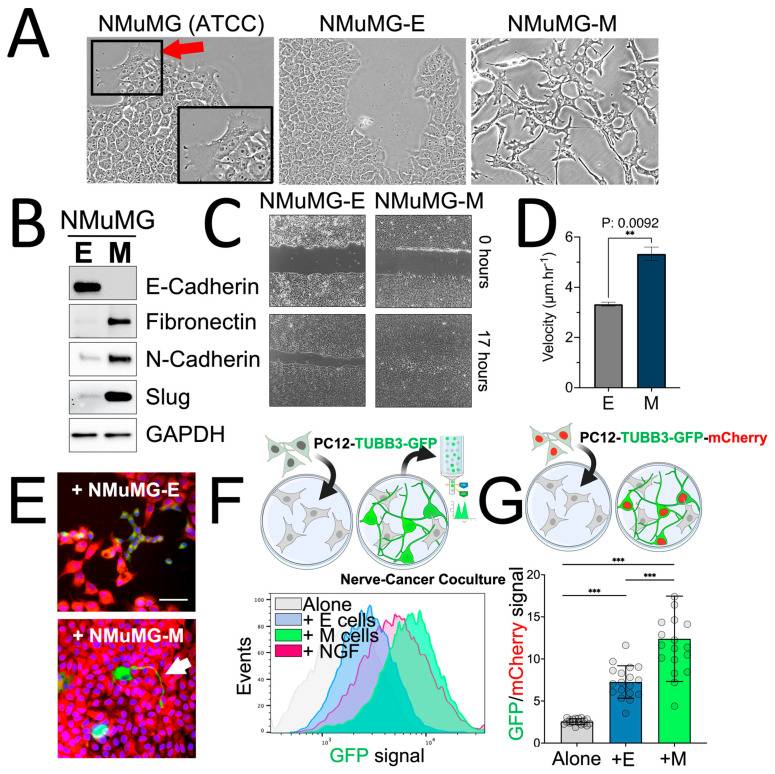
Breast cancer plasticity promotes neuron precursor axonogenesis. (**A**) To test the model developed for studying cancer-induced neuronal differentiation, we developed a model of breast cancer plasticity and epithelial-mesenchymal transition (EMT). The NMuMG cell cultures from ATCC are mostly epithelial cells but are heterogenous cultures having mixed phenotypes, with a few cells exhibiting mesenchymal morphologies (red arrows). NMuMG clones with epithelial (NMuMG-E) and mesenchymal (NMuMG-M) morphologies were isolated by single-cell seeding and regrowing of isogenic culture having epithelial or mesenchymal characteristics. (**B**) Western blot analysis of the regenerated culture confirms the occurrence of an EMT, consisting of the expression of the epithelial marker E-cadherin in NMuMG-E cells that disappears in NMuMG-M cells and the acquisition of the expression of mesenchymal markers fibronectin, N-cadherin, and Slug in NMuMG-M cells (Original Western blot, see Appendix A). GAPDH serves as a loading control. (**C**) Wound healing assay shows a greater velocity of migration in NMuMG-M cells compared to NMuMG-E cells, confirming the biological impact of the EMT on the cells’ aggressive behaviors. (**D**) To verify the relevance of NMuMG E/M to reflect the cell EMT-related plasticity functionally, we performed wound healing assays and quantified them by live-cell microscopy with the workflow published by Jonkman and colleagues [47]. (**E**) Fluorescence microscopy imaging of PC12-TUBB3-GFP cells in coculture with NMuMG-E or NMuMG-M constitutively expressing the mCherry fluorophore (red Channel), show increased GFP signal with the NMuMG-M cells, consistent with the development of extended neurites (White arrow). (**F**) PC12-TUBB-GFP cells were cultivated as unstimulated monoculture (Alone), incubated in direct coculture with either epithelial (+E cells) or mesenchymal (+M cells) cells, or treated with NGF (100 ng/mL) (+NGF) for 5 days as a positive control for differentiation. Flow cytometry revealed a moderate induction of PC12 neuronal precursor differentiation under NMuMG-E stimulation but a robust induction of differentiation under stimulation with NMuMG-M, where neuronal cells developed even more differentiation than those stimulated with NGF. (**G**) Cocultures of PC12-TUBB3-GFP-mCherry (constitutively expressing mCherry fluorophore) mixed with NMuMG-E or NMuMG-M cells were analyzed using an automated fluorescent plate reader. The mCherry expression allowed normalization of the assay by reflecting quantities of PC12 cells in the wells, and the mCherry/GFP ratio was used as an indicator of neuronal differentiation. Statistics: unpaired Student *t*-test ** *p* < 0.01; *** *p* < 0.001. Scale bar: 50 µm.

## Data Availability

Data and biological materials are available upon reasonable request to S.G.

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
