# Peer review of "A CRISPR/Cas9-Based Assay for High-Throughput Studies of Cancer-Induced Innervation"

_cancers, 2023, doi:10.3390/cancers15072026_

Round 1

Reviewer 1 Report

This is an interesting paper. The experimental design is sound and data interpretation is appropriate. However the following points should be addressed.

1. The authors have used NMuMG cells as "cancer" cells - however they just seem to be immortalized mammary gland cells? https://www.atcc.org/products/crl-1636. These cells are tumorigenic, but it produces benign cystadenomas. Therefore, it is unclear if these kinds of cells are the right choice to show the effect of neuronal differentiation in relation to cancer?

2. In the introduction, the authors state “The mecha-68 nisms in which cancers promote their own innervation remain mostly unknown”. This is an over-simplification that simply ignores the many manuscripts that have highlighted mechanisms by which cancer cells promote tumor innervation through the secretion of neurotrophic growth factors such as NGF. At minimum the following recent research paper PMID: 34785777 and the following review PMID: 30944117 should be cited

3. What are the intermediate differentiation states of N2A-TUBB3-GFP cells in response to the stimulation with NGF up to 10 days? Is it similar to those of PC12-TUBB3-GFP cells (Figure 3)? If not, why? Although figure 4A showed the GFP expression increase after 5 days of NGF treatment, but it is interesting to see whether there is kinetic difference between 2 cell lines. 

4. Can NMuMG-E and NMuMG-M control N2A differentiation?

5. Can the authors explain why co-culture with NMuMG-M PC12-TUBB3-GFP showed higher differentiation than NGF treatment (Figure. 5F)?

6. Any statistic difference in Figure 5G? If not, can the authors explain the reasons?

Reviewer 2 Report

Cancer innervation has become accepted as a new hallmark of cancer. Nerves innervating cancer tissue can stimulate cancer growth, as evidenced by findings that increased nerve density in cancer tissue is associated with more aggressive cancer growth. This stimulatory effect is mediated by neurotransmitters released from these nerves in the cancer microenvironment. On the other hand, the cancer microenvironment itself stimulates the ingrowth of new axons into the cancer tissue, creating a vicious circle. Therefore, approaches that reduce the innervation of cancer tissue or interfere with the interactions between neurons and cancer may provide a new tool for cancer treatment. However, our understanding of the processes and mechanisms related to the innervation of cancer tissue is still only partial.

In their manuscript Galappaththi et al. describe a new approach useful for more precisely studying the process related to cancer innervation. Described assays may be useful for quantifying cancer-induced axonogenesis in different cancer cell lines, and thus also for determining the efficacy of different therapeutic approaches based on modulating nerve-cancer interactions. Importantly, their essay can be used to study the role of plasticity of cancer cells in promoting cancer innervation.

Since the methodology used is original and appropriate and the data obtained are original, I fully support the publication of the paper.

Comments:

Please state in the Introduction that the cancer tissue can also induce a change in the phenotype of the neurons that innervated the original tissue (e.g., switching from a sensory neuron phenotype to a sympathetic neuron phenotype).

Round 2

Reviewer 1 Report

The authors have adequately addressed the issues raised in the initial review.